# Comparison the effects of carotid endarterectomy with carotid artery stenting for contralateral carotid occlusion

Yaxuan Sun[1]*, Yongxia Ding[2], Kun Meng[1], Bin Han[1], Jing Wang[1], Yan Han[1]

1 Department of Neurology, Shanxi Provincial People's Hospital, Taiyuan, Shanxi, China, 2 College of Nursing, Shanxi Medical University, Shanxi, China

* rvlm@163.com

## Abstract

### Background

There have been inconsistent results regarding the use of carotid artery endarterectomy (CEA) versus carotid artery stenting (CAS) for contralateral carotid occlusion (CCO). This study aimed to determine the optimal revascularization technique for patients with CCO.

### Methods

We systematically searched the PubMed, Embase, and Cochrane Library databases to identify eligible studies published from inception to January 2, 2021. Odds ratios (ORs) with 95% confidence intervals (CIs) were used to calculate pooled effect estimates using a random-effects model. Sensitivity, subgroup, and publication bias analyses were also performed.

### Results

Six studies involving 6,953 patients were selected for inclusion in this meta-analysis. Our results showed that while CEA was not associated with an increased risk of stroke compared to CAS (OR: 1.07; 95% CI: 0.75–1.51; P = 0.713), CEA was associated with a reduced risk of death compared to CAS (OR: 0.45; 95% CI: 0.29–0.70; P < 0.001). Furthermore, there were no significant differences between CEA and CAS for the risks of myocardial infarction (OR: 1.38; 95% CI: 0.73–2.62; P = 0.319) or major adverse cardiovascular events (OR: 1.03; 95% CI: 0.56–1.88; P = 0.926). Finally, the risk of myocardial infarction for CEA versus CAS was affected by disease status, while the risk of major adverse cardiovascular events was affected by the proportions of patients with male gender, coronary artery disease, and current or prior smoking.

### Conclusion

This study found that CEA and CAS resulted in similar outcomes for patients with CCO, while the risk of death was reduced in patients treated with CEA. Further high-level evidence should be collected to verify the results of this study.

**Data Availability Statement:** All relevant data are within the paper and its Supporting information files.

**Funding:** The author(s) received no specific funding for this work.

**Competing interests:** The authors have declared that no competing interests exist.

## Introduction

Contralateral carotid occlusion (CCO) accounts for nearly 5%–15% of carotid artery stenosis cases and has been demonstrated to be an independent risk factor for carotid endarterectomy (CEA) [1–4]. CEA is considered a gold-standard surgical technique for prevention of stroke in patients with severe stenosis or occlusion; however, the perioperative and long-term effects of CEA in patients with severe contralateral carotid stenosis or occlusion are variable [5–8]. The North American Symptomatic Carotid Endarterectomy Trial found that, though CEA was superior to medical management alone, CCO patients had a relatively high risk of perioperative stroke after CEA [9, 10], potentially because atheromatous plaques at carotid bifurcations are removed using CEA. Moreover, CEA has been shown to improve cerebral perfusion and to reduce the risk of stroke via washout of cerebral emboli from border-zone areas [11–13].

More recently, carotid artery stenting (CAS) has been introduced as an alternative treatment strategy for patients with carotid stenosis or CCO, especially for CCO patients with contraindications to CEA [14]. However, several studies have found that hemodynamic disturbances are associated with an increased risk of periprocedural stroke after CAS, always occurring within 6 hours of the procedure [15–18]. Previous systematic reviews and meta-analyses have been performed to compare the treatment effects of CEA versus CAS for patients with CCO [19, 20]. However, these two previous studies pooled only a small number of studies and were unable to clarify whether the treatment effects of CEA differed from CAS based on patient characteristics. Therefore, we performed a systematic review and meta-analysis to compare the effects of CEA versus CAS in patients with CCO.

## Methods

### Data sources, search strategy, and selection criteria

The Preferred Reporting Items for Systematic Reviews and Meta-Analysis Statement was used to guide the performing and reporting of this systematic review and meta-analysis [21]. Studies comparing the effects of CEA versus CAS in patients with CCO were considered potentially eligible for inclusion in this meta-analysis. The PubMed, Embase, and Cochrane Library databases were systematically searched for articles published from inception to January 2, 2021 using the following search terms: ("occlusion" AND "carotid" AND "contralateral") AND ("endarterectomy" OR "carotid artery stenting"). The reference lists of retrieved studies were also manually reviewed to identify any other eligible studies meeting the inclusion criteria.

Studies were included if they met the following criteria: (1) Patients: CCO; (2) Intervention: CEA; (3) Control: CAS; (4) Outcomes: stroke, death, myocardial infraction, and major adverse cardiovascular events; and (5) Study design: no restrictions on study design, including randomized controlled trials and prospective or retrospective cohort studies. The above study selection process was independently performed by two reviewers, and any disagreements were settled by group discussion until a consensus was reached.

### Data collection and quality assessment

Two reviewers abstracted data and performed quality assessments for each study, and conflicts between reviewers were resolved by an additional reviewer with reference to the original article. The following items were abstracted from each study: first author's name, publication year, country, study design, sample size, mean age, male proportion, symptomatic patients, prior transient ischemic attacks, prior stroke, coronary artery disease, hypertension, diabetes mellitus, current or prior smoking, imbalance characteristics between CEA and CSA groups, and reported outcomes. The quality of individual studies was assessed by the Newcastle-Ottawa

Scale (NOS), a comprehensive and validated tool for assessing the quality of observational studies in meta-analyses [22]. The so-called "starring system" for each study ranged from 0–9. Studies with 7–9 stars were regarded as having high quality and those with 4–6 stars were regarded as having moderate quality.

## Statistical analysis

The effects of CEA versus CAS on the risks of stroke, death, myocardial infarction, and major adverse cardiovascular events were calculated on the basis of the events that occurred and the sample sizes in each group. Odds ratios (OR) with 95% confidence intervals (CI) were then calculated using a random-effects model [23, 24]. The $I^2$ and Q statistic were applied to assess the heterogeneity across included studies, with significant heterogeneity defined as an $I^2$ of > 50.0% or a P-value of < 0.10 [25, 26]. The robustness of the pooled conclusions was assessed using a sensitivity analysis with sequential excluding of single studies [27]. Subgroup analyses were performed based on age, male gender, disease status, coronary artery disease, hypertension, diabetes mellitus, and smoking, and the differences between subgroups were assessed using an interaction P test [28]. Both qualitative and quantitative methods were applied to assess for potential publication bias, including funnel plots and Egger's and Begg's tests [29, 30]. The P-values for pooled conclusions were two-sided, and the inspection level was 0.05. Statistical analyses were conducted using STATA software (version 10.0; Stata Corporation, College Station, TX, USA).

## Results

### Search of the published literature

An initial electronic search identified 1,354 articles, but 411 of these articles were excluded due to duplication of titles. In addition, 907 studies were excluded because of reporting on irrelevant topics. The remaining 36 studies were retrieved for full-text evaluations, and 30 of these studies were excluded because of no appropriate control (n = 18, other disease status (n = 9), or a review (n = 3). Reviewing the reference lists of relevant studies identified one other potential study, and this study was identified in an electronic search. Finally, a total of six studies were selected for the final meta-analysis [2, 31–35], with details regarding the literature search and study selection process shown in Fig 1.

### Study characteristics

All included studies had a retrospective cohort design, and these studies involved a total of 6,953 patients with CCO. The baseline characteristics of the included studies are shown in Table 1. Four studies were conducted in the USA, one study was conducted in Italy, and the remaining study was conducted in Korea. The sample sizes ranged from 57–4,326 patients, and a total of 2,423 patients had presented with symptomatic CCO. Study quality was assessed using the NOS and the detail results are shown in Table 2, with three studies having six stars and the remaining three studies having five stars.

### Stroke

After pooling all included studies, there was no significant difference between CEA and CAS for the risk of stroke (OR: 1.07; 95% CI: 0.75–1.51; P = 0.713; Fig 2), and no evidence of heterogeneity was observed ($I^2$ = 0.0%; P = 0.498). The sensitivity analysis found the pooled conclusion was stable (S1 Fig). Subgroup analyses found no significant differences between CEA and

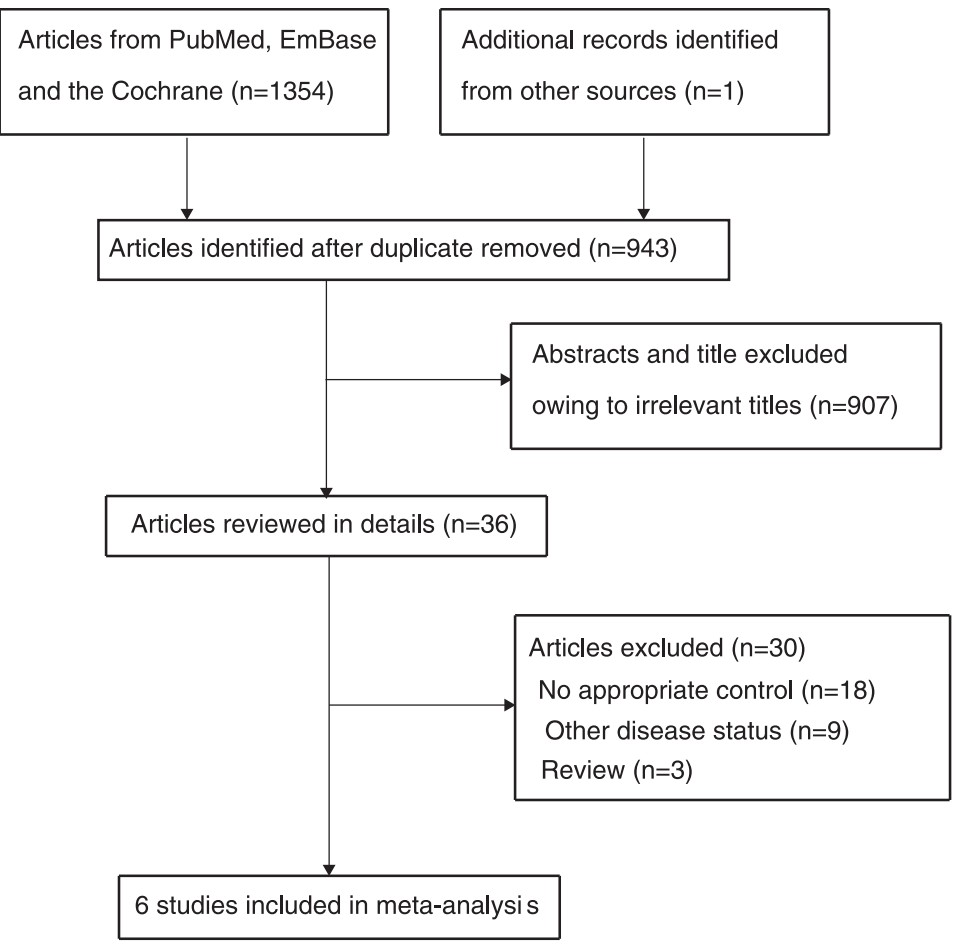

**Fig 1. Literature search and study selection process.**

CAS for the risk of stroke in any subgroup (Table 3). No significant publication bias for stroke was detected (P-value for Egger's test: 0.511; P-value for Begg's test: 1.000; S2 Fig)

## Death

Four studies reported the effects of CEA versus CAS on the risk of death, and the summary OR indicated that CEA was associated with a reduced risk of death compared to CAS (OR: 0.45; 95% CI: 0.29–0.70; P < 0.001; Fig 3). There was no evidence of heterogeneity across included studies ($I^2$ = 0.0%; P = 0.491). Sensitivity analysis indicated no significant difference between CEA and CAS for the risk of death when removing the study conducted by Nejim et al., which specifically included a large number of patients and reported a higher incidence of death (S3 Fig). Subgroup analyses found that CEA was more protective than CAS for the risk of death in the following subgroups: average age < 70.0 years, male proportion ≥ 70.0%, coronary artery disease proportion < 40.0%, hypertension proportion ≥ 80.0%, diabetes mellitus proportion ≥ 30.0%, and smoking proportion ≥ 40.0% (Table 3). There was no significant publication bias for death (P-value for Egger's test: 0.422; P-value for Begg's test: 0.734; S4 Fig).

**Table 1. Baseline characteristics of included studies and patients.**

| Study | Country | Study design | Sample size | Mean age (years) | Male (%) | Symptomatic patients (%) | Prior TIA (%) | Prior stroke (%) | CAD (%) | Hypertension (%) | DM (%) | Smoking (%) | Imbalance characteristics (CEA/CAS) |
|---|---|---|---|---|---|---|---|---|---|---|---|---|---|
| Brewster 2012 [31] | USA | Retrospective | 57 (18/39) | 67.7 | 70.2 | 49.1 (8/20) | 36.8 | 35.1 | 14.0 | 87.7 | 36.8 | 33.3 | Prior neck surgery (5.6%/46.1%) |
| Faggioli 2013 [2] | Italy | Retrospective | 75 (37/38) | 70.9 | 70.7 | 33.3 (13/12) | NA | NA | 30.7 | 92.0 | 26.7 | 10.7 | Age (65.0/76.6 years), CAD (20.0%/43.2%), CRF (5.9%/24.3%) |
| Ricotta 2014 [32] | USA | Retrospective | 1,794 (666/1,128) | 69.9 | 63.5 | 48.7 (266/607) | 22.0 | 30.1 | 56.7 | 85.7 | 34.0 | NA | Symptomatic (39.9%/53.8%), CAD (52.7%/59.0%), CHF (8.4%/14.2%), diabetes (31.1%/35.7%), COPD (18.3%/22.6%), cancer (11.7%/15.1%) |
| Yang 2014 [33] | Korea | Retrospective | 94 (44/50) | 64.8 | 89.7 | 35.1 (10/24) | 20.6 | 12.4 | 47.4 | 69.1 | NA | 62.9 | Symptomatic (22.7%/48.0%), stroke (2.3%/22.0%), CAD (61.4%/38.0%) |
| Nejim 2017 [34] | USA | Retrospective | 4,326 (3,274/1,052) | 68.0 | 71.0 | 28.3 (790/433) | NA | 47.6 | 30.8 | 89.0 | 34.5 | 85.0 | CHF (9.6%/15.6%), COPD (16.2%/20.3%), stroke (56.4%/24.0%), Symptomatic (46.8%/60.8%) |
| Turley 2019 [35] | USA | Retrospective | 607 (565/42) | 70.0 | 70.0 | 39.5 (NA) | NA | NA | NA | 52.9 | 28.7 | 34.9 | Baseline ipsilateral carotid stenosis (mild [1.1%/2.4%]; moderate [32.7%/16.7%; severe [66.2%/81.0%]]) |

CAD, coronary artery disease; CHF: congestive heart failure; COPD: chronic obstructive pulmonary disease; CRF: chronic renal failure; DM, diabetes mellitus; TIA, transient ischemic attack

**Table 2. Quality scores of included studies using Newcastle-Ottawa Scale.**

| Study | Selection | | | | Comparability | Outcome | | | NOS |
|---|---|---|---|---|---|---|---|---|---|
| | Representativeness of the exposed cohort | Selection of the non exposed cohort | Ascertainment of exposure | Demonstration that outcomes was not present at start of study | Comparability on the basis of the design or analysis | Assessment of outcome | Adequate follow-up duration | Adequate follow-up rate | Overall score |
| Brewster 2012 [31] | 0 | 1 | 1 | 1 | 1 | 1 | 0 | 0 | 5 |
| Faggioli 2013 [2] | 0 | 1 | 1 | 1 | 1 | 1 | 0 | 1 | 6 |
| Ricotta 2014 [32] | 1 | 1 | 1 | 1 | 1 | 1 | 0 | 0 | 6 |
| Yang 2014 [33] | 0 | 0 | 1 | 1 | 1 | 1 | 0 | 1 | 5 |
| Nejim 2017 [34] | 1 | 1 | 1 | 1 | 1 | 1 | 0 | 0 | 6 |
| Turley 2019 [35] | 0 | 0 | 1 | 1 | 1 | 1 | 1 | 0 | 5 |

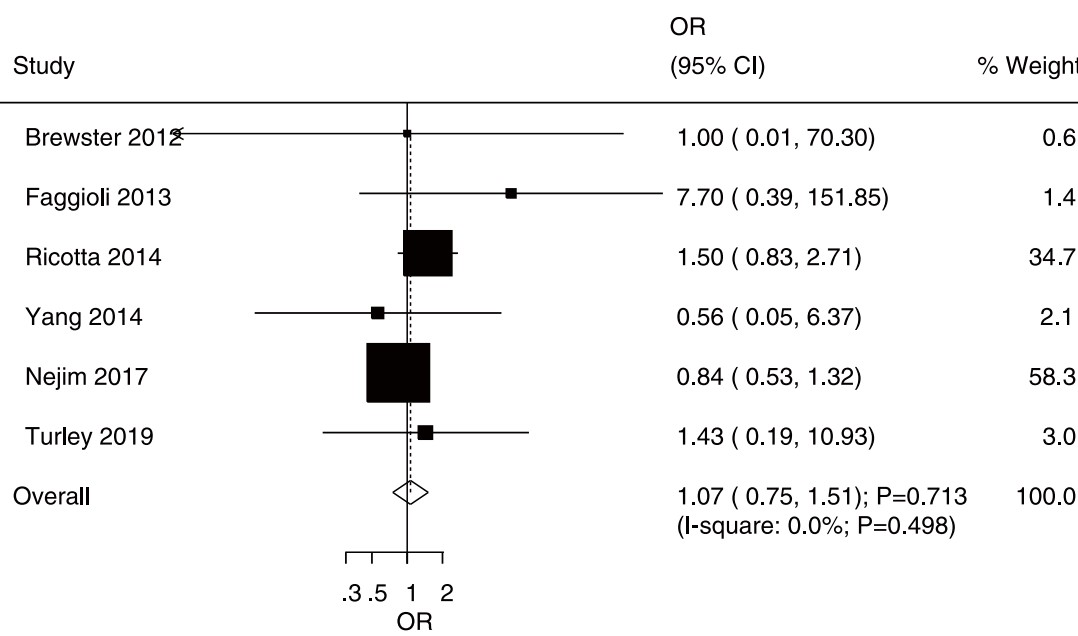

**Fig 2. Effects of carotid endarterectomy (CEA) versus Carotid Artery Stenting (CAS) on the risk of stroke.**

### Myocardial infarction and major adverse cardiovascular events

The number of studies evaluating the effects of CEA versus CAS on the risks of myocardial infarction or major adverse cardiovascular events were three and three, respectively (Fig 4). There were no significant differences between CEA and CAS for the risk of myocardial infarction (OR: 1.38; 95% CI: 0.73–2.62; P = 0.319) or major adverse cardiovascular events (OR: 1.03; 95% CI: 0.56–1.88; P = 0.926). There was no evidence of heterogeneity for myocardial infarction ($I^2$ = 0.0%; P = 0.941); however, there was potentially significant heterogeneity for major adverse cardiovascular events ($I^2$ = 69.0%; P = 0.040). Subgroup analyses found that CEA versus CAS was not associated with the risk of myocardial infarction in any subgroup and that treatment effect differences on the risk of myocardial infarction between CEA and CAS were significantly affected by the disease status (P = 0.048). Moreover, CEA was associated with a reduced risk of major adverse cardiovascular events in subgroups with a male proportion of ≥ 70.0%, a coronary artery disease proportion of < 40.0%, and a smoking proportion of ≥ 40.0% (Table 3). Furthermore, the proportions of males (P = 0.015), patients with coronary artery disease (P = 0.040), and patients with current or prior smoking (P = 0.013) significantly affected the treatment effects of CEA versus CAS for the risk of major adverse cardiovascular events.

### Discussion

Studies have already proposed using CAS as an alternative treatment strategy to CEA in CCO patients [36, 37]; however, inconsistent treatment results have been obtained in these studies. In this meta-analysis, a total of 6,953 patients with CCO from six retrospective studies were included, and these patients had a broad variety of characteristics. This study found no significant differences between CEA and CAS for the risk of stroke, myocardial infarction, or major adverse cardiovascular events in patients with CCO. However, CEA was associated with a reduced risk of death compared to CAS. The protective role of CEA versus CAS on the risk of

**Table 3. Subgroup analyses for stroke, death, myocardial infarction, and major adverse cardiovascular events.**

| Outcomes | Factors | Groups | Number of studies | OR (95% CI) | P-value | $I^2$ (%) | $P_{Q\ statistic}$ | P-value between subgroups |
|---|---|---|---|---|---|---|---|---|
| Stroke | Age (years) | ≥ 70.0 | 2 | 2.43 (0.46–13.00) | 0.298 | 0.0 | 0.360 | 0.325 |
| | | < 70.0 | 4 | 1.03 (0.72–1.47) | 0.877 | 0.0 | 0.464 | |
| | Male (%) | ≥ 70.0 | 5 | 0.89 (0.58–1.37) | 0.601 | 0.0 | 0.658 | 0.163 |
| | | < 70.0 | 1 | 1.50 (0.83–2.71) | 0.179 | - | - | |
| | Disease status | Asymptomatic | 3 | 1.29 (0.68–2.43) | 0.432 | 0.0 | 0.850 | 0.077 |
| | | Symptomatic | 3 | 0.58 (0.24–1.42) | 0.231 | 4.4 | 0.351 | |
| | CAD (%) | ≥ 40.0 | 2 | 1.42 (0.80–2.52) | 0.233 | 0.0 | 0.439 | 0.428 |
| | | < 40.0 | 3 | 0.93 (0.49–1.77) | 0.830 | 3.6 | 0.354 | |
| | Hypertension (%) | ≥ 80.0 | 4 | 1.14 (0.69–1.89) | 0.610 | 25.3 | 0.260 | 0.904 |
| | | < 80.0 | 2 | 0.97 (0.21–4.60) | 0.972 | 0.0 | 0.561 | |
| | DM (%) | ≥ 30.0 | 3 | 1.06 (0.70–1.61) | 0.783 | 13.6 | 0.314 | 0.544 |
| | | < 30.0 | 2 | 2.43 (0.46–13.00) | 0.298 | 0.0 | 0.360 | |
| | Smoking (%) | ≥ 40.0 | 2 | 0.83 (0.53–1.30) | 0.411 | 0.0 | 0.747 | 0.193 |
| | | < 40.0 | 3 | 2.18 (0.45–10.44) | 0.331 | 0.0 | 0.615 | |
| Death | Age (years) | ≥ 70.0 | 1 | 3.11 (0.12–79.15) | 0.492 | - | - | 0.239 |
| | | < 70.0 | 3 | 0.44 (0.28–0.68) | < 0.001 | 0.0 | 0.598 | |
| | Male (%) | ≥ 70.0 | 3 | 0.41 (0.26–0.67) | < 0.001 | 0.0 | 0.448 | 0.369 |
| | | < 70.0 | 1 | 0.70 (0.25–1.98) | 0.502 | - | - | |
| | Disease status | Asymptomatic | 2 | 0.73 (0.36–1.48) | 0.382 | 0.0 | 0.872 | 0.054 |
| | | Symptomatic | 2 | 0.50 (0.04–5.54) | 0.571 | 56.5 | 0.130 | |
| | CAD (%) | ≥ 40.0 | 1 | 0.70 (0.25–1.98) | 0.502 | - | - | 0.369 |
| | | < 40.0 | 3 | 0.41 (0.26–0.67) | < 0.001 | 0.0 | 0.448 | |
| | Hypertension (%) | ≥ 80.0 | 4 | 0.45 (0.29–0.70) | < 0.001 | 0.0 | 0.491 | - |
| | | < 80.0 | 0 | - | - | - | - | |
| | DM (%) | ≥ 30.0 | 3 | 0.44 (0.28–0.68) | < 0.001 | 0.0 | 0.598 | 0.239 |
| | | < 30.0 | 1 | 3.11 (0.12–79.15) | 0.492 | - | - | |
| | Smoking (%) | ≥ 40.0 | 1 | 0.40 (0.25–0.65) | < 0.001 | - | - | 0.525 |
| | | < 40.0 | 2 | 0.93 (0.08–11.42) | 0.956 | 11.0 | 0.289 | |
| MI | Age (years) | ≥ 70.0 | 0 | - | - | - | - | - |
| | | < 70.0 | 3 | 1.38 (0.73–2.62) | 0.319 | 0.0 | 0.941 | |
| | Male (%) | ≥ 70.0 | 2 | 1.31 (0.64–2.69) | 0.461 | 0.0 | 0.904 | 0.745 |
| | | < 70.0 | 1 | 1.70 (0.42–6.85) | 0.455 | - | - | |
| | Disease status | Asymptomatic | 2 | 0.72 (0.33–1.59) | 0.415 | 0.0 | 0.871 | 0.048 |
| | | Symptomatic | 2 | 5.51 (0.86–35.14) | 0.071 | 0.0 | 0.416 | |
| | CAD (%) | ≥ 40.0 | 1 | 1.70 (0.42–6.85) | 0.455 | - | - | 0.745 |
| | | < 40.0 | 2 | 1.31 (0.64–2.69) | 0.461 | 0.0 | 0.904 | |
| | Hypertension (%) | ≥ 80.0 | 3 | 1.38 (0.73–2.62) | 0.319 | 0.0 | 0.941 | - |
| | | < 80.0 | 0 | - | - | - | - | |
| | DM (%) | ≥ 30.0 | 3 | 1.38 (0.73–2.62) | 0.319 | 0.0 | 0.941 | - |
| | | < 30.0 | 0 | - | - | - | - | |
| | Smoking (%) | ≥ 40.0 | 1 | 1.32 (0.64–2.74) | 0.455 | - | - | 0.941 |
| | | < 40.0 | 1 | 1.00 (0.01–83.85) | 1.000 | - | - | |

(*Continued*)

**Table 3.** (Continued)

| Outcomes | Factors | Groups | Number of studies | OR (95% CI) | P-value | $I^2$ (%) | $P_{Q\,statistic}$ | P-value between subgroups |
|---|---|---|---|---|---|---|---|---|
| MACE | Age (years) | ≥ 70.0 | 1 | 1.13 (0.33–3.82) | 0.844 | - | - | 0.690 |
| | | < 70.0 | 2 | 1.02 (0.48–2.19) | 0.960 | 84.1 | 0.012 | |
| | Male (%) | ≥ 70.0 | 2 | 0.73 (0.54–0.99) | 0.045 | 0.0 | 0.470 | 0.015 |
| | | < 70.0 | 1 | 1.55 (0.92–2.61) | 0.099 | - | - | |
| | CAD (%) | ≥ 40.0 | 1 | 1.55 (0.92–2.61) | 0.099 | - | - | 0.040 |
| | | < 40.0 | 1 | 0.71 (0.52–0.97) | 0.034 | - | - | |
| | Hypertension (%) | ≥ 80.0 | 2 | 1.02 (0.48–2.19) | 0.960 | 84.1 | 0.012 | 0.690 |
| | | < 80.0 | 1 | 1.13 (0.33–3.82) | 0.844 | - | - | |
| | DM (%) | ≥ 30.0 | 2 | 1.02 (0.48–2.19) | 0.960 | 84.1 | 0.012 | 0.690 |
| | | < 30.0 | 1 | 1.13 (0.33–3.82) | 0.844 | - | - | |
| | Smoking (%) | ≥ 40.0 | 1 | 0.71 (0.52–0.97) | 0.034 | - | - | 0.013 |
| | | < 40.0 | 2 | 1.48 (0.91–2.38) | 0.111 | 0.0 | 0.640 | |

CAD, coronary artery disease; DM, diabetes mellitus; MACE, major adverse cardiovascular events; MI, myocardial infarction

death was mainly detected in subgroups with an average age of < 70.0 years, a male proportion of ≥ 70.0%, a coronary artery disease proportion of < 40.0%, a hypertension proportion of ≥ 80.0%, a diabetes mellitus proportion of ≥ 30.0%, and a smoking proportion of ≥ 40.0%. Interestingly, we also noted that CEA was associated with a reduced risk of major adverse cardiovascular events in subgroups with a male proportion of ≥ 70.0%, a coronary artery disease proportion of < 40.0%, and a smoking proportion of ≥ 40.0%. The results of this study support the use of CEA for patients with CCO. Furthermore, this study identifies specific populations who could benefit most significantly from CEA.

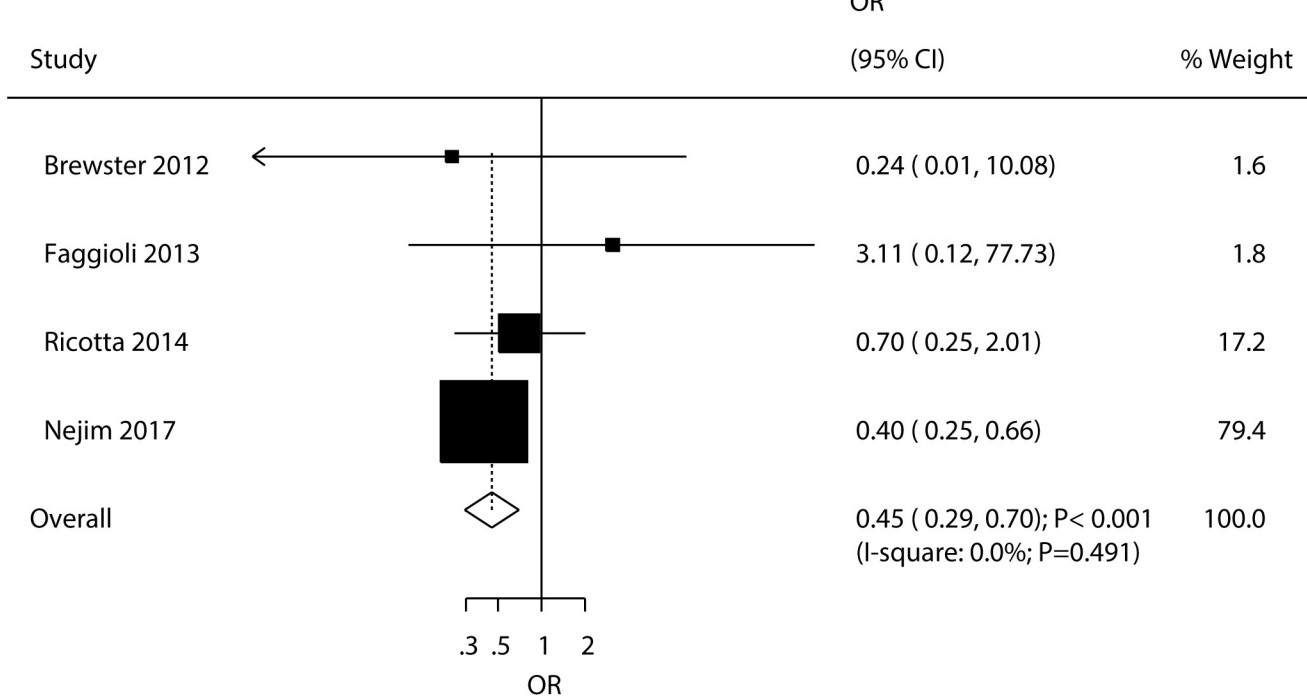

**Fig 3. Effects of carotid endarterectomy (CEA) versus Carotid Artery Stenting (CAS) on the risk of death.**

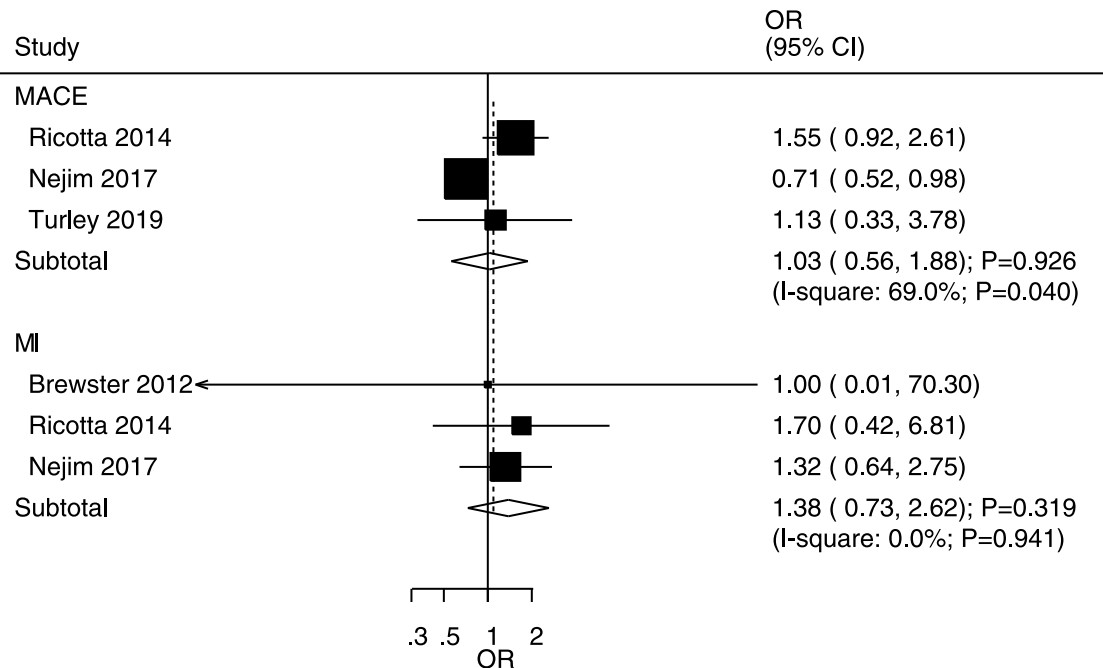

**Fig 4. Effects of carotid endarterectomy (CEA) versus Carotid Artery Stenting (CAS) on the risks of myocardial infarction and major adverse cardiovascular events.**

A similar prior meta-analysis included five retrospective studies and found that CEA was associated with a reduced risk of 30-day periprocedural mortality, while the risks of stroke, myocardial infarction, and major adverse cardiovascular events were similar in the CEA and CAS groups of CCO patients [19]. Another important meta-analysis conducted by Xin et al. identified four studies and found similar conclusions [20]. Moreover, these studies pointed out that the risk of transient ischemic attacks did not statistically differ between CEA and CAS. However, these studies could not identify whether the treatment effects of CEA versus CAS differed based on patient characteristics. Furthermore, newly published articles should be included in an updated meta-analysis, and pooled results should be reevaluated. Therefore, the present systematic review and meta-analysis was performed to compare the effects of CEA versus CAS in patients with CCO using the most current literature.

There were no significant differences between CEA and CAS for the risks of stroke, myocardial infarction, or major adverse cardiovascular events. Most of the included studies reported similar conclusions; however, the study conducted by Nejim et al. found that CEA was associated with a reduced risk of major adverse cardiovascular events in comparison with CAS [34]. The beneficial effect of CEA on the risk of major adverse cardiovascular events was mainly detected in symptomatic patients. Moreover, subgroup analyses found that CEA was associated with a reduced risk of major adverse cardiovascular events in the subgroups with a male proportion of $\geq 70.0\%$, a coronary artery disease proportion of $< 40.0\%$, and a smoking proportion of $\geq 40.0\%$. These findings may have resulted from the inclusion of the study conducted by Nejim et al. due to this study's large sample size and high weighting in the pooled results. Moreover, the definition of major adverse cardiovascular events differed across the included studies, which might have contributed to these results.

Similar to prior meta-analyses, we noted that CEA was associated with a reduced risk of death compared to CAS, with this significant benefit mainly observed in the subgroups of

patients with an average age of $< 70.0$ years, a male proportion of $\geq 70.0\%$, a coronary artery disease proportion of $< 40.0\%$, a hypertension proportion of $\geq 80.0\%$, a diabetes mellitus proportion of $\geq 30.0\%$, and a smoking proportion of $\geq 40.0\%$. These findings may result from CCO patients presenting with severe atherosclerosis and resultant soft plaques. In these patients, emboli can escape during CAS because this technique requires repeated passes through stenotic blood vessels and does not block blood flow. Moreover, a rejection reaction to CAS has been associated with an increased risk of mortality [38]. Moreover, the benefit effect of CEA on the risk of death mainly attribute the study conducted by Nejim et al [34], this study specifically reported the proportion of congestive heart failure, chronic obstructive pulmonary disease, and symptomatic patients in CEA group was lower than CAS group, which could explained the potential difference for the risk of death between CEA and CAS. Subgroup analyses suggested these populations could obtain more benefit from CEA than CAS in protecting against the risk of death.

Several strengths of this study should be highlighted: (1) the analysis of this study included data from a large number of patients, and the conclusions had more robustness than from any individual study; (2) there was no evidence of heterogeneity except for in major adverse cardiovascular events, which may have been explained by the various definitions used across the included studies; and (3) subgroup analyses were performed, and the results of these analyses identify specific populations who may obtain more benefit from CEA.

The limitations of this study should also be acknowledged. First, all of the included studies had a retrospective cohort design, and the characteristics of patients who underwent CEA versus CAS were not balanced, which could have affected the prognosis of these patients. Second, due to the smaller number of included studies, the results of subgroup analyses were variable and need further verification. Third, although no significant publication bias was observed, it is inevitable that some bias occurred due to the nature of an analysis based on published articles. Finally, the details analyses were restricted because pooled data from individual studies were used.

In conclusion, this study found that CEA and CAS contributed similar effects to the risks of stroke, myocardial infarction, and major adverse cardiovascular events, while CCO patients treated with CEA obtained more benefit for preventing death than those treated with CAS. Further randomized controlled trials should be performed to verify the findings of this study.

## Supporting information

**S1 Fig. Sensitivity analysis for stroke.**
(DOCX)

**S2 Fig. Funnel plot for stroke.**
(DOCX)

**S3 Fig. Sensitivity analysis for death.**
(DOCX)

**S4 Fig. Funnel plot for death.**
(DOCX)

**S1 Checklist.**
(DOC)

## Author Contributions

**Conceptualization:** Yaxuan Sun, Yan Han.

**Data curation:** Bin Han.

**Formal analysis:** Kun Meng.

**Investigation:** Kun Meng, Jing Wang.

**Methodology:** Yongxia Ding.

**Resources:** Bin Han.

**Software:** Yan Han.

**Writing – original draft:** Yaxuan Sun, Yongxia Ding.

**Writing – review & editing:** Yaxuan Sun.

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
