## [Decision Letter · Decision Letter 0]

19 Mar 2021

PONE-D-21-04598

Comparison the effects of carotid endarterectomy with carotid artery stenting for contralateral carotid occlusion

PLOS ONE

Dear Dr. Sun

Thank you for submitting your manuscript to PLOS ONE. After careful consideration, we feel that it has merit but does not fully meet PLOS ONE’s publication criteria as it currently stands. Therefore, we invite you to submit a revised version of the manuscript that addresses the points raised during the review process.

Please add a table with the assessment of the quality of the included studies. 

Please submit your revised manuscript in 60 days. If you will need more time than this to complete your revisions, please reply to this message or contact the journal office at plosone@plos.org. Please include the following items when submitting your revised manuscript:

We look forward to receiving your revised manuscript.

Kind regards,

Timir Paul

Academic Editor

PLOS ONE

Journal Requirements:

3. Please include captions for *all* your Supporting Information files at the end of your manuscript, and update any in-text citations to match accordingly. Please see our Supporting Information guidelines for more information: http://journals.plos.org/plosone/s/supporting-information

Reviewers' comments:

Reviewer's Responses to Questions

**Comments to the Author**

1. Is the manuscript technically sound, and do the data support the conclusions?

Reviewer #1: Yes

Reviewer #2: Yes

2. Has the statistical analysis been performed appropriately and rigorously? 

Reviewer #1: I Don't Know

Reviewer #2: Yes

3. Have the authors made all data underlying the findings in their manuscript fully available?

Reviewer #1: Yes

Reviewer #2: Yes

4. Is the manuscript presented in an intelligible fashion and written in standard English?

Reviewer #1: Yes

Reviewer #2: Yes

5. Review Comments to the Author

Reviewer #1: Nice written work on very important and contested problem. I wish they show the criteria of patients selected for CEA vs CAS in the studies. I would also like to know why CEA associated with lower mortality. Could it be that patients declined for CEA because of higher surgical risk go for CAS and ending up having more mortality ? Can the authors comment on that if possible.

Reviewer #2: Overall well written article by Sun et al. comparing the effects of carotid endarterectomy with carotid artery stenting for

contralateral carotid occlusion in a total of 6953 patients. The study found that CEA and CAS had similar outcomes for

patients with CEA versus carotid stenting, while the risk of death was reduced in patients treated with CEA. Overall, these conclusions are similar to a previous meta analysis on the topic by Texakalidis et al. in 2018 involving 6346 patients. The authors report there sample size is larger after addition of an addition retrospective study. However, I do not feel this study adds much to the scientific literature given the additional studies including in this analysis were all retrospective in nature and not RCTs. There is only a difference of approximately 600 patients in the sample size between these two analyses and as such, the similar findings are not surprising.

6. PLOS authors have the option to publish the peer review history of their article (what does this mean?). If published, this will include your full peer review and any attached files.

Reviewer #1: No

Reviewer #2: No

---

## [Author Response · Author response to Decision Letter 0]

5 Apr 2021

Dear editor,

Thanks for your kind reply, and thanks for the reviewers’ beneficial comments. We have revised the manuscript according to the reviewers’ comments and the point to point response is enclosed to this file. The reply was noted with tracked.

Response to editor

Question 1: Please add a table with the assessment of the quality of the included studies.

Response: Thanks for this suggestion, and the quality scores of included studies using Newcastle-Ottawa Scale have already presented in Table 2. 

Question 2: Please ensure that your manuscript meets PLOS ONE's style requirements, including those for file naming. The PLOS ONE style templates can be found at

Response: Thanks for this suggestion, and the manuscript have already updated according to PLOS ONE's style requirements. All of changes in the revised manuscript with tracked.

Question 3: Please review your reference list to ensure that it is complete and correct. If you have cited papers that have been retracted, please include the rationale for doing so in the manuscript text, or remove these references and replace them with relevant current references. Any changes to the reference list should be mentioned in the rebuttal letter that accompanies your revised manuscript. If you need to cite a retracted article, indicate the article’s retracted status in the References list and also include a citation and full reference for the retraction notice.

Response: Thanks for this suggestion, and the reference lists have already updated according to the PLOS ONE's requirements. 

Question 4: Please include captions for *all* your Supporting Information files at the end of your manuscript, and update any in-text citations to match accordingly. Please see our Supporting Information guidelines for more information: http://journals.plos.org/plosone/s/supporting-information

Response: Thanks for this suggestion, and the captions for supporting information have already added at the end of the manuscript. Moreover, the in-text citations have already updated with tracked.

Response to reviewer #1

General comments: Nice written work on very important and contested problem. I wish they show the criteria of patients selected for CEA vs CAS in the studies. I would also like to know why CEA associated with lower mortality. Could it be that patients declined for CEA because of higher surgical risk go for CAS and ending up having more mortality ? Can the authors comment on that if possible.

Response: We appreciate the reviewer given this constructive comments. We have already added the characteristics between CEA and CAS groups in Table 1 if they associated with statistically significant. Moreover, the results regarding the high risk of death in CAS group have already discussed in Discussion section. 

Response to reviewer #2

General comments: Overall well written article by Sun et al. comparing the effects of carotid endarterectomy with carotid artery stenting for contralateral carotid occlusion in a total of 6953 patients. The study found that CEA and CAS had similar outcomes for patients with CEA versus carotid stenting, while the risk of death was reduced in patients treated with CEA. Overall, these conclusions are similar to a previous meta analysis on the topic by Texakalidis et al. in 2018 involving 6346 patients. The authors report there sample size is larger after addition of an addition retrospective study. However, I do not feel this study adds much to the scientific literature given the additional studies including in this analysis were all retrospective in nature and not RCTs. There is only a difference of approximately 600 patients in the sample size between these two analyses and as such, the similar findings are not surprising.

Response: Thanks for this suggestion. We appreciate the reviewer given this constructive comments. We acknowledge the results of this study were consistent with the study conducted by Texakalidis et al. However, the current study also assessed whether the treatment effects of CEA versus CAS are differing based on age, male gender, disease status, coronary artery disease, hypertension, diabetes mellitus, and smoking. We have already addressed this question in Discussion section. 

Sincerely,

Yaxuan Sun

---

## [Editor Report · Decision Letter 1]

12 Apr 2021

Comparison the effects of carotid endarterectomy with carotid artery stenting for contralateral carotid occlusion

PONE-D-21-04598R1

Dear Dr. Sun,

We’re pleased to inform you that your manuscript has been judged scientifically suitable for publication and will be formally accepted for publication once it meets all outstanding technical requirements.

Kind regards,

Timir Paul

Academic Editor

PLOS ONE

---

## [Editor Report · Acceptance letter]

21 Apr 2021

PONE-D-21-04598R1 

Comparison the effects of carotid endarterectomy with carotid artery stenting for contralateral carotid occlusion 

Dear Dr. Sun:

I'm pleased to inform you that your manuscript has been deemed suitable for publication in PLOS ONE. Congratulations! Your manuscript is now with our production department. 

Kind regards, 

on behalf of

Dr. Timir Paul 

Academic Editor

PLOS ONE